# Effects of Methyl Jasmonate Fumigation on the Growth and Detoxification Ability of *Spodoptera litura* to Xanthotoxin

**DOI:** 10.3390/insects14020145

**Published:** 2023-01-31

**Authors:** Lina Chen, Jia Song, Jun Wang, Mao Ye, Qianqian Deng, Xiaobao Wu, Xiaoyi Wu, Bing Ren

**Affiliations:** 1The Provincial Key Laboratory for Agricultural Pest Management of the Mountainous Region, Institute of Entomology, Guizhou University, Guiyang 550005, China; 2State Key Laboratory Breeding Base of Green Pesticide and Agricultural Bioengineering, Guizhou University, Guiyang 550025, China; 3Key Laboratory of Green Pesticide and Agricultural Bioengineering, Ministry of Education, Guizhou University, Guiyang 550025, China; 4Guiyang Plant Protection and Quarantine Station, Guiyang 550081, China

**Keywords:** methyl jasmonate, *Spodoptera litura*, growth, detoxifying ability

## Abstract

**Simple Summary:**

Methyl jasmonate (MeJA) is a volatile substance derived from jasmonic acid (JA), and it responds to biotic and abiotic stresses by participating in interplant communication. Because MeJA is hydrophobic and it can easily move between plant cells, it is considered a propagable form of JA, which can utilize this method to induce plant defenses. Pests are able to ‘eavesdrop’ on plant signal molecules as cues for activating the detoxification system to protect themselves from plant defenses, and insects have evolved a variety of mechanisms to metabolize, sequester, or detoxify plant toxins. *Spodoptera litura* is an omnivorous insect that damages a variety of crops worldwide. We analyzed the growth of *S. litura* and its detoxification ability against xanthotoxin after exposure to different concentrations of MeJA fumigation, and the effective role of MeJA in inducing insects’ defense response to the toxin was studied. We demonstrated that MeJA is effective at inducing *S. litura* defense response by increasing its detoxifying enzyme activities, but the enhanced detoxifying ability could not overcome the strong toxins.

**Abstract:**

Methyl jasmonate (MeJA) is a volatile substance derived from jasmonic acid (JA), and it responds to interbiotic and abiotic stresses by participating in interplant communication. Despite its function in interplant communication, the specific role of MeJA in insect defense responses is poorly understood. In this study, we found that carboxylesterase (CarE) activities, glutathione-S-transferase (GSTs) activities, and cytochrome mono-oxygenases (P450s) content increased more after the feeding of diets containing xanthotoxin, while larvae exposed to MeJA fumigation also showed higher enzyme activity in a dose-dependent manner: lower and medium concentrations of MeJA induced higher detoxification enzyme activities than higher concentrations of MeJA. Moreover, MeJA improved the growth of larvae fed on the control diet without toxins and diets with lower concentrations of xanthotoxin (0.05%); however, MeJA could not protect the larvae against higher concentrations of xanthotoxin (0.1%, 0.2%). In summary, we demonstrated that MeJA is effective at inducing *S. litura* defense response, but the enhanced detoxifying ability could not overcome the strong toxins.

## 1. Introduction

Plants and herbivorous insects share a contradictory but everlasting relationship [1]; insects recognize host plants according to specific chemical clues while plants have evolved diverse strategies to survive and adapt to herbivory [2]. To cope with the herbivores, plants invest their energy in the production of various defensive chemicals, which are known as plant secondary metabolites [3,4]. These metabolite have the ability to aid plant defenses against insect pests through defense signaling pathways [5,6]. The secondary metabolites of plants such as phenols, alkaloids, and terpenes can directly inhibit herbivore growth and development or are even lethal to insects [7]. It has been reported that *Spodoptera litura* exposed to phytochemicals showed reduced survival and prolonged development [8]. Further, the pupal weights in *Spodoptera exigua* were significantly lower when they were fed on the high iridoidglycosides line [9]. In addition, some host plants, such as groundnuts, show higher phenolic induction when attacked by *Helicoverpa armigera* [10]. However, in the evolution of plants and insects, herbivores have been able to metabolize, isolate, and even detoxify plant toxins [11,12,13]. Herbivores are also able to use plant-signaling molecules to activate their own detoxification functions by ‘eavesdropping’, through which they can protect themselves from various plant defenses [14,15]. A recent study found that insects could modify plants’ secondary metabolites by detoxifying enzymes to neutralize plant toxins [16]. Some insects have even acquired the ability to horizontally transfer plant-derived genes to help them to circumvent the resistance traits of host plants [17]. Due to complex diets, the detoxification abilities among various kinds of insects are different, and omnivorous herbivores possess more detoxification gene families than monophagous herbivores [18].

The detoxification enzymes, including cytochrome P450 enzymes (P450s), carboxylesterase (CarE), and glutathione-S-transferases (GSTs), play important roles in insect adaptation and the detoxification defense against plant toxins [19,20,21]. These three detoxification enzymes in phytophagous insects can decompose lipophilic exogenous toxins into hydrophilic intrinsic substances and are widely used to detoxify secondary metabolites [22]. It was reported that the GST and P450 enzyme activities of *Lymantria dispar* larvae were significantly increased after consuming artificial feed containing plant secondary metabolites [12]. When the American white moth, *Hyphantria cunea*, feeds on the host plant with the highest flavonoid content, its CarE and P450 enzyme activities are the highest [23]. The GSTs of *Anopheles gambiae* and *Nilaparvata lugens* are also involved in the detoxification process of insecticides [24,25], indicating that the CarE, GST, and P450 enzymes play important roles in insect adaptation and the detoxification defense against plant secondary metabolites [20,21]. Additionally, although xanthotoxin, a linear furanocoumarin, is a phytotoxin, numerous studies have shown that it is able to induce detoxification enzyme activities in herbivores [26,27,28]. For example, the expression of P450 genes in the *Spodoptera litura* larvae was upregulated in response to xanthotoxin [29].

The volatile organic compounds (VOCs) released by herbivores attacking plants are attractive to arthropod predators and parasitoids [30]. Interestingly, plants can also make use of VOCs released by herbivore-attacked neighbors to activate defenses before being attacked [31]. Phytohormone jasmonate (JA) and its derivatives are important plant hormones that mediate plant development and defend against herbivorous insects [32]. The JA pathway is usually related to the response of feeding of chewing herbivores and regulates the secondary metabolites to prevent feeding or inhibit the production of digestion, as well as to induce plant volatiles that can repel herbivores and attract their natural enemies [33,34,35,36]. However, herbivores have evolved multiple ways to avoid, resist, or manipulate plant defenses [37,38,39]. Zhang et al. (2016) showed that the peach aphid and corn earworm can directly use the defense response signal substance, JA or SA, as a signal molecule before the host plant produces secondary metabolites by upregulating the expression of related detoxification genes and salivary gland-related genes in vivo [15]. MeJA was originally identified in the jasmine flower of *Jasminum grandiflorum* and it is thought to be a volatile form of JA, which is a substance that is widely distributed in various plants [40]. MeJA is hydrophobic and it can easily move between plant cells as it is considered a propagable form of JA, which can utilize this method to induce plant defenses [41]. In addition, MeJA application can increase the production of a broad spectrum of toxic allelochemicals and proteinase inhibitors to repel herbivorous insects [42,43]. However, despite its function in interplant communication, the specific role of MeJA in insect defense responses is poorly understood. The tobacco cutworm, *Spodoptera litura* (Fabricius), is a major widespread and broadly polyphagous pest that feeds on a variety of important economic crops, such as tomato, cotton, soybean, tobacco, and peanut [44]. In this study, we analyzed the growth of *S. litura* and its detoxification ability against xanthotoxin after exposure to different concentrations of MeJA fumigation, and the effective role of MeJA in inducing insects’ defense responses to the toxin was studied.

## 2. Materials and Methods

### 2.1. Insect Rearing

The population of *S. litura* was obtained from the Institute of Entomology, Guizhou University, China, and the larvae were raised on artificial feed in the laboratory. Feed formulations included soybean powder (40 g), yeast extract (40 g), wheat bran (40 g), ascorbic acid (3.2 g), casein (8 g), agar (8 g), sorbic acid (0.8 g), cholesterol (0.08 g), choline chloride (2 g), and water (400 mL) [45]. The larvae were reared at 27 °C under a light/dark photoperiod of 16:8 h and in 60% relative humidity. All samples were laboratory-bred offspring, and the rearing and feeding conditions of captive individuals were consistent. The 4th-instar larvae of *S. litura* were used for the experiments.

### 2.2. Methyl Jasmonate Fumigation Treatment

MeJA (≥98%, Sigma-Aldrich) was dissolved by 5% ethanol and was then configured to 20 µg/µL, 200 µg/µL, and 2000 µg/µL MeJA solutions using distilled water. The 4th-instar larvae of *S. litura* were transferred to a plastic cup (75 mm × 60 mm × 27 mm) containing an artificial diet, then 2 μL of each concentration of MeJA was added to the absorbent cotton fixed on the lid of the cup with a pipette, then an equal amount of 2 μL ethanol solution was added dropwise as control, and then the cup lid was quickly closed for MeJA exposure for 24 h.

### 2.3. Xanthotoxin Treatment

Xanthotoxin (≥98%, Yuanye, Shanghai, China) was dissolved in dimethyl sulfoxide (DMSO) and the xanthotoxin solution was stirred evenly with an artificial feed to prepare the xanthotoxin feed with concentrations of 0.05%, 0.1%, and 0.2%. These concentrations were based on previous studies on the relationship between xanthotoxin and insect detoxification [46,47]. The 4th-instar larvae of *S. litura* were fed on diets with different concentrations of xanthotoxin.

### 2.4. Determination of CarE, GSTs Activity and P450s Contents

*S. litura* larvae from 24 h 0.05% xanthotoxin and 24 h MeJA treatments were dissected on a PBS buffer. Midguts and fat body from five larvae were pooled to represent a sample. Midguts and fat body were homogenized in a precooled homogenization buffer (136.89 mM NaCl, 2.67 mM KCl, 8.1 mM Na_2_HPO_4_, 1.76 mM KH_2_PO_4_, pH 7.4) and were centrifuged at 8000× *g* for 10 min at 4 °C. Next, the supernatant was collected for enzymatic determination. Detection kits for the GSTs and CarE activity were bought from Beijing Solarbio Science & Technology Co., Ltd. Beijing, China. One unit (U) of CarE activity was defined as each increase of 1 in catalytic absorbance value per minute at 37 °C; one unit (U) of the GSTs’ activity was defined as 1 µmol of 1-chloro-2,4-dinitrobenzene (CDNB) conjugated with reduced glutathione (GSH) per minute at 37 °C; cytochrome P450s’ contents were detected by an ELISA assay kit, which were bought from Nanjing Jiancheng Bioengineering Institute, China. The total protein content was measured by the Bradford method following the kit instructions (Comin Biotech, Suzhou, China). All assays for enzyme activities were performed with 5 replications.

### 2.5. Developmental Parameters

The fourth-instar larvae were pre-exposed to different concentrations of MeJA as described above for 24 h and were then transferred to new cups with diets containing 0.05%, 0.1%, or 0.2% xanthotoxin or diets without toxin. The initial weight of each larva was recorded. All larvae were weighed again on the third day after transfer to toxic chow to determine weight gain. We observed daily larva survival and developmental stages until all larvae underwent pupation or died. In addition, the pupae were weighed when the larvae were pupated after 24 h. The experiment was conducted on 60 larvae with 3 replications of each concentration (20 larvae per replicate).

### 2.6. Statistics and Analysis

The SPSS 21.0 statistical package for Windows was used for the statistical analysis. Differences in enzyme activities between MeJA- and xanthotoxin-treated and untreated larvae were determined using the Student’s t-test. The differences between the developmental parameters among the treatments were evaluated by a one-way and a factorial ANOVA at *P* = 0.05 using the Duncan’s post hoc test.

## 3. Results

### 3.1. Increased Activities of Detoxifying Enzymes and P450 Contents in S. litura Larvae after Feeding on Xanthotoxin Diet

To determine the effect of xanthotoxin on *S. litura*, the enzyme activities of the larvae were analyzed (Figure 1). When fed on the diet containing 0.05% xanthotoxin, the CarE and GST activities in the midguts of larvae were 1.3-fold (Figure 1A) and 2.0-fold (Figure 1C) compared to the control larvae fed on a diet without the toxin, respectively. In the fat body, there was no significant difference in GST activity after eating the feed containing xanthotoxin (Figure 1D) while the P450 activity in the midguts of the larvae showed no significant difference from those fed on the non-toxin diet (Figure 1E). In the fat body, the CarE activity and P450 contents were induced about 1.7-fold (Figure 1B) and 1.8-fold (Figure 1F) by feeding on the diet containing 0.05% xanthotoxin relative to the control larvae, respectively. These results suggest that the CarE, GST activities, and P450 contents in the *S. litura* larvae significantly increased in response to the xanthotoxin.

### 3.2. S. litura Larvae Exposed to MeJA Showed Higher Detoxifying Enzyme Activities

We exposed the *S. litura* larvae to different concentrations of MeJA for 24 h and then analyzed the CarE, GST activities, and P450 contents in the midguts and fat body of the larvae. The results showed varying effects of the different concentrations of MeJA on the activities of the three tested enzymes. After exposure to the medium concentration of MeJA (200 µg/µL), the CarE activity in the midguts of the larvae significantly induced, by 2.8-fold, relative to that of the control larvae, while the CarE activity in the midguts exposed to lower (20 µg/µL) and higher (2000 µg/µL) concentrations of MeJA did not differ from that in the control larvae (Figure 2A). In the fat body, the larvae exposed to 2000 µg/µL MeJA showed 1.6-fold CarE activity than the control larvae, while the CarE activity in the larvae exposed to the lower and medium concentrations of MeJA did not differ from that of the controls (Figure 2B). The GSTs’ activity in the midgut of the larvae exposed to lower and higher concentrations of MeJA were 3.0-fold and 5.7-fold, respectively, than the control larvae (Figure 2C), while only the larvae exposed to lower concentrations of MeJA showed higher GST activity in their fat body relative to the control (Figure 2D). For the P450 contents, the larvae exposed to the medium (200 µg/µL) and high (2000 µg/µL) concentrations of MeJA showed 1.1-fold and 1.5-fold, respectively, relative to the control in the midguts (Figure 2E) and 1.9-fold and 2.6-fold contents, respectively, relative to the control in the fat body (Figure 2F).

### 3.3. Enhanced Detoxification Ability of S. litura after Exposure to Different Concentrations of MeJA

The *S. litura* larvae pre-exposed to different concentrations of MeJA were fed on diets containing 0.05%, 0.1%, and 0.2% xanthotoxin and diets without toxins. The two-way ANOVA showed that the effects of the MeJA treatment, the xanthotoxin treatment, and their interactions on weight gain, growth rate, prepupal instar, pupal stage, and pupal weight were significant (Table A1).

The *S. litura* larvae fed on the diet without toxins performed better than those fed on the toxic diet (Figure 3A,B). The toxic effect of xanthotoxin increased with the concentration and the weight gain of the *S. litura* larvae not pre-exposed to MeJA decreased 28.45%, 43.77%, and 68.93% after being fed on diets containing 0.05%, 0.1%, and 0.2% xanthotoxin, respectively (Figure 3A). Similarly, the pupal weight decreased by 11.35%, 18.85%, and 42.45% (Figure 3B) and durations of the 4th, 5th, and 6th were prolonged (Table 1).

Interestingly, the MeJA treatment improved the growth of the larvae without toxin stress. When fed on a diet without the toxin, the larvae pre-exposed to 200 µg/µL of MeJA exhibited a 19.85% higher weight gain compared to the control larvae without MeJA exposure (Figure 3A). When fed on the 0.05% xanthotoxin diet, the larvae pre-exposed to 20 µg/µL of MeJA showed a 42.18% higher weight gain and 8.03% higher pupal weights than the control larvae without MeJA exposure, respectively (Figure 3A,B). The duration of 4th-instar and the prepupal instar of the larvae were shortened, but the durations of the 5th- and 6th-instars were unaffected (Table 1). These results suggest that 20 µg/µL of MeJA could help the larvae to adapt to 0.05% xanthotoxin. However, the MeJA could not alleviate the toxic effect of the xanthotoxin on the larvae when they were fed on the 0.1% and 0.2% xanthotoxin diets. When fed on the 0.1% xanthotoxin diet, the larvae pre-exposed to 200 µg/µL of MeJA showed only a 22.19% higher weight gain than the control larvae (Figure 3A). Additionally, the weight gain and pupal weight of *Spodoptera litura* larvae decreased significantly when fed with a high concentration (0.2%) toxic feed (Figure 3A,B). The duration of the 4th-, 5th-, and 6th-instar and the prepupal instar of the larvae were prolonged (Table 1). The pupal weight increased significantly compared with the control when the larvae fumigated with MeJA after feeding on the nontoxic feed (Figure 3B). After feeding with a 0.1% feed, the pupal weight of *S. litura* was significantly increased by 5.63% with a high concentration (2000 µg/µL) of MeJA fumigation compared with the control, while other concentrations had no significant difference. MeJA fumigation had no effect on pupal weight when feeding a 0.5% feed. When fed on the 0.2% xanthotoxin diet, larvae pre-exposed to 20 µg/µL MeJA showed 6.88% higher pupal weights than those in the control larvae (Figure 3B).

## 4. Discussion

Plants and insects have co-evolved for about 400 million of years. The never-ending arm race between these two entities allows them to circumvent the defense mechanisms of each other. In addition to plants’ defense against insects, insects’ antidefense mechanisms are already common in plant-insect interactions [48]. In response to plant defenses, herbivorous insects have evolved a range of antidefense mechanisms, such as detoxification enzymes, including CarE, GSTs, and P450s [49,50,51]. CarE belongs to the major superfamily of α/β hydrolase proteins that catalyze the hydrolysis of carboxylate esters to alcohols and carboxylic acids [52]. GSTs are involved in the detoxification of endogenous and exogenous compounds; they are a common enzyme family in aerobic organisms [53]. In addition, they are also involved with intracellular transportation, hormone biosynthesis, and protection against oxidative stress, which play an important role in insect resistance [54]. Cytochrome P450 mono-oxygenase is the largest and most diverse enzyme superfamily; it is considered the main detoxification enzyme in insect resistance to plant allelochemicals [55]. Yang et al. (2017) found that phenolics induce GSTs or CarE activity in insects in the interaction of ferulic acid with glutathione S-transferase and carboxylesterase genes of the brown planthopper [54]. In the study of locusts, it was also proved that insects could produce a large amount of detoxification enzymes to deal with the damage of plant secondary metabolites. Huang et al. (2021) treated the short star-winged locust with quercetin and found that its CYP450 and GST activities increased [56]. GST and P450 activities significantly increased in the *Lymantria dispar* larvae after feeding on quercetin-treated artificial diets [12]. It was reported that *CYP6B8* could metabolize various phytochemicals in the *Helicoverpa zea*, such as xanthotoxin and quercetin [57]; these results suggest that herbivorous insects increase their detoxification enzyme activities in the presence of plant allelochemicals [55,58]. In addition, insects can use these detoxification enzymes to inactivate a variety of toxins, including xanthotoxin [59,60,61]. Here, our results showed that CarE, GST activities, and P450 contents significantly increased in the *S. litura* larvae in response to xanthotoxin stress, and the three enzymes showed different activities in different tissues. Elevated activity of CarEs were expressed in the midgut and fat body of larvae after 24 h feeding; GSTs’ activity increased higher in the midgut while the P450s’ contents increased higher in the fat body (Figure 1). These results suggest that the three detoxification enzymes play active roles against xanthotoxin. The different induction effects on the three enzymes in different tissue may depend on their detoxifying functions to the toxin. In addition to the midgut tissue, the fat body is also a site of detoxifying enzyme metabolism [53,62,63].

Plant hormones, such as jasmonic acid (JA) and salicylic acid (SA), play important roles in regulating the signaling network of plant defenses against insect pests [64]. However, Li et al. (2002) found that both lower and higher concentrations of JA (2.9 or 290 μg g^−1^) and SA (12 μg g^−1^ or 12,000 μg g^−1^) could induce the expression of P450 genes (*CYP6B8*, *CYP6B28*, *CYP6B9,* and *CYP6B27*) in the midgut and fat body of corn earworms (*Helicoverpa zea*), and they pointed out that the ability to use plant signal molecules as cues for activating a detoxification system may be of particular value to a broadly polyphagous herbivore [14]. Additionally, herbivore-induced plant volatiles (HIPVs) can considerably reduce caterpillar susceptibility to insecticide through induction-enhanced detoxification mechanisms. Luo et al. (2022) exposed *S. litura* larvae to volatiles from *S. litura*-damaged tomato plants. They found elevated activities of GSTs, P450s, and esterases (ESTs), and a higher expression of related genes in contrast to those exposed to volatiles from undamaged tomato plants, and these larvae were significantly less susceptible to the insecticides [65]. MeJA and methyl salicylate (MeSA) could act as airborne signaling molecules between damaged and undamaged neighboring plants [66]. Fan et al. (2014) have compared the effects of the feeding of MeJA and the exposure to MeJA fumigation on the growth and detoxifying abilities of *Helicoverpa armigera* larvae. They also found that larvae that experienced MeJA fumigation exhibited the highest P450 activity and increased detoxifying abilities in *H. armigera* against cotton defense responses induced by MeJA; moreover, MeJA fumigation did not exhibit the growth of *H. armigera* [19]. In our study, we exposed *S. litura* larvae to different concentrations of MeJA fumigation for 24 h. The results showed that the different concentrations of MeJA had different inducing effects on CarE, GSTs’ activities, and P450s’ contents: the lower and medium concentrations of MeJA induced higher detoxification-enzyme activity than the higher concentration of MeJA. Moreover, the MeJA exposure alleviated the toxic effects of the xanthotoxin on the larvae, the larvae pre-exposed to MeJA showed higher weight gain and pupal-weight, and the durations of the 5th- and 6th-instars were curtailed when certain concentrations were applied (Figure 3, Table 1). Our results collectively indicate that the broadly polyphagous herbivorous insects could ‘intercept’ plant communication to prepare for the upcoming danger.

Plants attacked by insects can perceive volatile organic compounds and warn neighboring plants to strengthen their defenses; plants growing close to herbivore-infested conspecifics expressed higher levels of resistance to herbivores than plants that grew farther away [67]. Our previous study also found that MeJA fumigation significantly induced neighboring tomato plants’ resistance, while the inductive effects of MeJA fumigation depended on its concentration and operating distance [68]. However, in nature, insects could freely move for feeding, allowing them to choose the most suitable host plants [69,70]. We have shown here that MeJA exposure improved the growth of larvae fed on a control diet without toxins and a diet with a lower concentration of xanthotoxin (0.05%), suggesting that larvae pre-exposed to MeJA could equip themselves before the accumulation of toxic concentrations of plant defence compounds and those larvae that experienced MeJA fumigation may benefit significantly when they transfer to feed on other plants with lower levels of defense. Thus, the adaptability of insects must be considered when jasmonic acid and its derivatives are applied to induce plant defense in the field.

## Figures and Tables

**Figure 1 insects-14-00145-f001:**
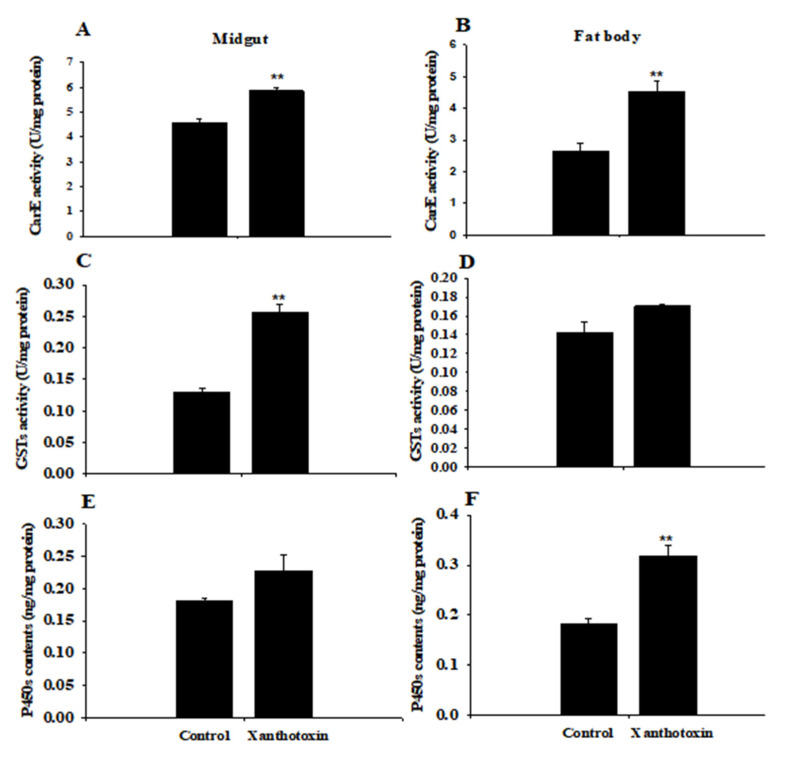
CarE (**A**,**B**), GST (**C**,**D**) activities, and P450 (**E**,**F**) contents in the midguts and fat body of *Spodoptera litura* larvae after feeding on a diet containing 0.05% xanthotoxin and a diet without toxin (control). Values are means ± SE (*n* = 5). Asterisks indicate significant differences between enzyme activities of larvae fed on diets with or without xanthotoxin, respectively (** *p*  <  0.01, Student’s *t*-test).

**Figure 2 insects-14-00145-f002:**
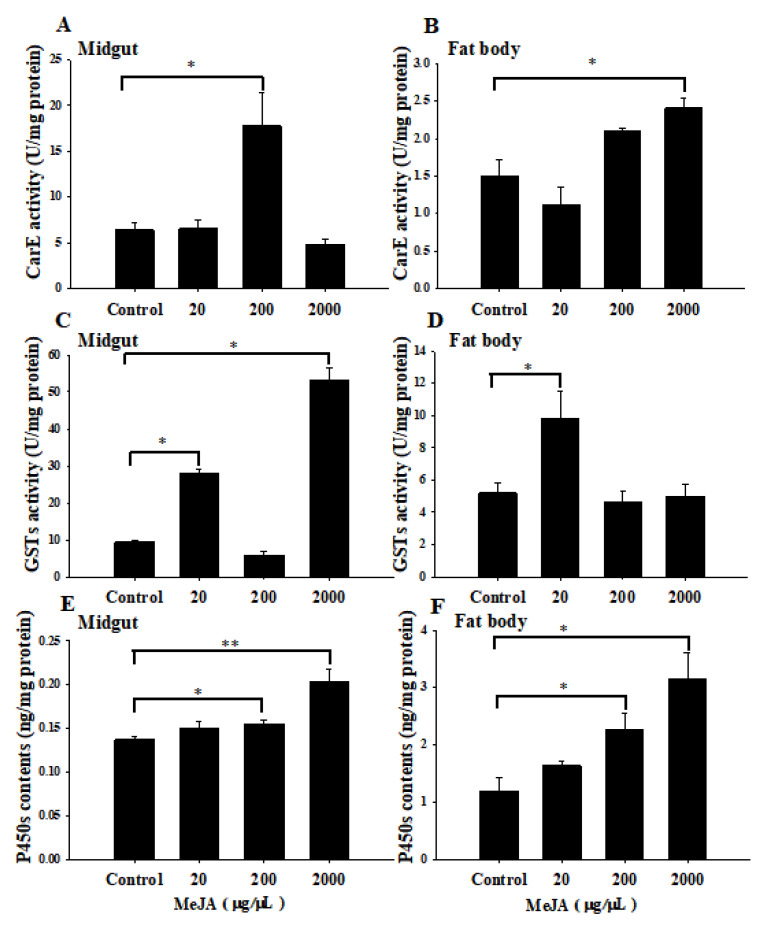
CarE (**A**,**B**), GST (**C**,**D**) activities and P450 (**E**,**F**) contents in midguts and fat body of *Spodoptera litura* larvae among different treatments. Values are means ± SE (*n* = 5). Asterisks indicate significant differences in enzyme activity in larvae exposed to different concentrations of MeJA compared to the controls (* *p*  <  0.05, ** *p*  <  0.01, Student’s *t*-test).

**Figure 3 insects-14-00145-f003:**
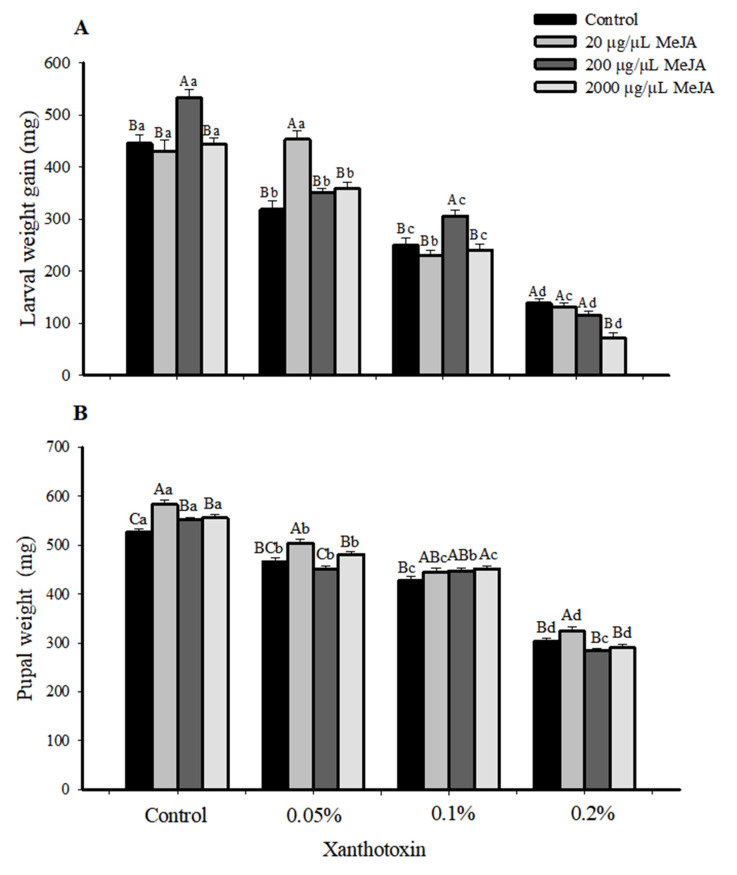
Effects of MeJA on growth of *Spodoptera litura* larvae under xanthotoxin stress. (**A**) weight gain and (**B**) pupal weight of *Spodoptera litura* larvae fed on xanthotoxin after fumigation with methyl jasmonate. Values are means ± SE (*n* = 60). The capital letters indicate the significant differences among different concentrations of MeJA in the same xanthotoxin concentrations; lowercase indicates the significant differences among different concentrations of xanthotoxin at the same MeJA concentrations. One-way ANOVA (Duncan’s, *p* < 0.05).

**Table 1 insects-14-00145-t001:** Effects of MeJA on developmental time of *Spodoptera litura* larvae under xanthotoxin stress. Values are means ± SE (*n* = 60). The capital letters in the same column indicate the significant differences among different concentrations of xanthotoxin at the same MeJA concentrations; the lowercases indicate the significant differences among different concentrations of MeJA at the same xanthotoxin concentrations. One-way ANOVA (Duncan’s, *p* < 0.05).

MeJA Concentration (μg/μL)	Xanthotoxin Concentration (%)	Duration of Fourth Instar (d)	Duration of Fifth Instar (d)	Duration of Sixth Instar (d)	Duration of Prepupal Instar (d)	Duration of Pupal Instar (d)
Control	0	3.10 ± 0.05 Bb	2.08 ± 0.10 Ca	1.98 ± 0.10 Aa	1.12 ± 0.04 Bc	9.71 ± 0.08 Ba
	0.05	3.05 ± 0.04 Bb	2.10 ± 0.11 Cb	2.27 ± 0.13 Ab	1.18 ± 0.06 Bb	9.70 ± 0.08 Ba
	0.1	3.12 ± 0.10 Bb	2.37 ± 0.10 Ba	1.49 ± 0.07 Bc	1.28 ± 0.06 Bb	10.00 ± 0.08 Aa
	0.2	3.37 ± 0.06 Ab	2.83 ± 0.05 Aa	2.25 ± 0.10 Ab	1.58 ± 0.08 Aa	10.13 ± 0.06 Ab
20	0	3.73 ± 0.06 Aa	1.67 ± 0.08 Bc	1.86 ± 0.07 Ca	1.97 ± 0.07 Aa	9.25 ± 0.06 Cc
	0.05	3.23 ± 0.07 Ba	1.60 ± 0.09 Bc	1.87 ± 0.07 Cc	1.77 ± 0.06 Ba	9.69 ± 0.08 Ba
	0.1	3.77 ± 0.05 Aa	2.00 ± 0.10 Ab	2.39 ± 0.09 Ba	1.27 ± 0.06 Db	10.08 ± 0.08 Aa
	0.2	3.75 ± 0.10 Aa	2.13 ± 0.08 Ac	2.87 ± 0.09 Aa	1.59 ± 0.06 Ca	10.07 ± 0.07 Ab
200	0	2.14 ± 0.04 Cc	2.39 ± 0.09 ABb	1.42 ± 0.07 Db	1.61 ± 0.08 Bb	9.10 ± 0.08 Cc
	0.05	2.23 ± 0.07 Cc	2.57 ± 0.09 Aa	2.68 ± 0.13 Ba	1.90 ± 0.09 Aa	9.69 ± 0.07 Ba
	0.1	2.69 ± 0.06 Bc	2.22 ± 0.07 Bab	1.88 ± 0.08 Cb	1.25 ± 0.06 Cb	9.24 ± 0.06 Bc
	0.2	3.04 ± 0.13 Ac	2.58 ± 0.07 Ab	3.04 ± 0.04 Aa	1.60 ± 0.07 Ba	10.14 ± 0.06 Ab
2000	0	2.05 ± 0.03 Ac	1.07 ± 0.03 Dd	2.09 ± 0.07 Ba	2.04 ± 0.09 Aa	9.51 ± 0.08 Cb
	0.05	2.06 ± 0.03 Ad	1.39 ± 0.07 Cc	2.24 ± 0.09 BCb	1.82 ± 0.05 Ba	9.76 ± 0.07 Ba
	0.1	2.12 ± 0.04 Ad	1.66 ± 0.07 Bc	2.42 ± 0.09 Ba	1.64 ± 0.07 Ba	9.71 ± 0.07 Bb
	0.2	2.12 ± 0.04 Ad	2.02 ± 0.05 Ac	2.87 ± 0.06 Aa	1.75 ± 0.06 Ba	10.43 ± 0.06 Aa

## Data Availability

All relevant data are within the paper.

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
