# Peer review of "Effects of Methyl Jasmonate Fumigation on the Growth and Detoxification Ability of Spodoptera litura to Xanthotoxin"

_insects, 2023, doi:10.3390/insects14020145_

Round 1
Reviewer 1 Report
The findings of study is quite interesting, however, some suggesations and comments are raised please see the file attached.

Author Response
Reviewer #1:
Point 1:You may improve your introduction by taking information from this paper
“Volatile signals from guava plants prime defense signaling and increase jasmonate-dependent herbivore resistance in neighboring citrus plants”
Insect detoxification enzymes more details are available here. Toxicity and enzyme inhibition activities of the essential oil and dominant constituents derived from Artemisia absinthium L. against adult Asian citrus psyllid Diaphorina …
Response 1 :Thanks, we have modified the introduction.
Point 2:Fumigant toxicity and biochemical properties of (α+ β) thujone and 1, 8-cineole derived from Seriphidium brevifolium volatile oil against the red imported fire ant Solenopsis Fumigation timing should be more than 48 h. Mostly 72 h after fumigation seal will be open to ensure efficacy.
Response 2 :Thanks, but we exposed S. litura to MeJA fumigation for 24 h,and found that detoxifying enzymes increased significantly and the larvae performanced more well than controls, it may be interesting to compare inductive effects of MeJA among different treated times.
Point 3:I have not found a positive control data in the experiment. for comparison purpose add positive control results along with the table and figures.
Response 3 :we take larvae exposed to equal amount of ethanol, which we used to dissolve MeJA, as positive control.
Point 4:What will be the cost benefit ration of jasmonic acid if we going to recommend it for fumigation e.g you can compare to the aluminum phosphide
Response 4 : sorry, we did not quite understand your question. We have compared the cost benefit of MeJA in another experiment, the results showed that plant benefit more from MeJA.
Point 5:The findings of the study is interesting , what will be the future recommendation regrinding the finding of study?
Response 5 :It is worth continuing to consider and study why herbivores have the specific molecular mechanism of detoxification. In addition, the effect of volatiles on herbivores should be considered while selecting the selection of plant volatiles to manage pests and improve plant defense.
Response to the comment: Thanks to the reviewers for discussing our experimental methods, our experimental objective was to see the detoxification enzyme activity of Spodoptera litura. 24 hours was shown to induce detoxification enzyme activity,this is very interesting idea to fumigation at different times. As for our positive control problem, I would like to explain that our experimental controls are only for the untreated samples, and the untreated samples are our experimental controls. It is worth continuing to consider and study why herbivores have the specific molecular mechanism of detoxification. In addition, the effect of volatiles on herbivores should be considered while selecting the selection of plant volatiles to manage pests and improve plant defense.
Reviewer 2 Report
Dear authors
Their study is interesting and applicate; however, the methodology and results presentation need to be improved to best understand. (see document attached)

Author Response
Reviewer #2:
Point 1:what is early? <6h, <12h <24h <48h?????
Response 1 : Sorry, our article didn't write clearly, we actually used the fourth instar larvae.
Point 2:the title is not clear (please write a title that have implicit the contend. (see results)
Response 2 : Thanks, we have modified in the title.
Point 3:write the methodology of forma sequential accord to the results for understand best, please.
Response 3 :Thanks, we have modified.
Point 4:why the control have various concentratiosn??? specific in the methodology
Response 4 :Have been modified clearly as described in Method 2.5
The fourth-instar larvae were pre-exposed to different concentrations of MeJA as described above for 24 hours and were then transferred to new cups with diets containing 0.05%, 0.1% or 0.2% xanthotoxin or diets without toxin.
Point 5:these concetrations??? MeJA 20 ug/uL ok but these concentrations???? explain in the metodology please.
Response 5 :We have explained in Methods.
Response to the comment:Thanks to the reviewers for their comments, we have revised the format questions, see the manuscript for details. Concerning the reviewer's on the early, ur article didn't write clearly, we actually used the fourth instar larvae. And have been modified clearly as described in Method 2.5. The fourth-instar larvae were pre-exposed to different concentrations of MeJA as described above for 24 hours and were then transferred to new cups with diets containing 0.05%, 0.1% or 0.2% xanthotoxin or diets without toxin. For the concentration problem, we need to explain that the concentration of methyl jasmonate is calculated using its density. And the MeJA (≥98%, Sigma-Aldrich) were dissolved by 5% ethanol, and were then configured to 20-, 200- and 2000-microgram/microliter MeJA solutions using distilled water.
Reviewer 3 Report
Comments to authors:
The manuscript by Chen et al. first exposed Spodoptera litura larvae to methyl jasmonate and then exposed to plant allelochemical Xanthotoxin to assess if is there any enhanced detoxification of xanthotoxin or not. The study is interesting; however, I find difficulty in understanding what the authors trying to say in the test because of so many typos, grammatical errors, and incomprehensible test.
Line 10: The full form of abbreviations needs to be provided when they are first used.
Line 17: Xanthotoxin is written as xanthoxytoxin; correct it.
Line No 19: Change “by increase” with “by increasing”.
Line 77: Change “detoxify ability” with “detoxification ability”.
Line 87: Delete “preparation” in this sentence.
Line 91: Delete “supplemented” in this sentence.
Materials and Methods should be written in the past tense as they have already been done.
Line 105: Dissolved xanthotoxin (≥98%, Yuanye, Shanghai, China) by dimethyl sulfoxide. In this sentence change “by” with “in.
Fig. 1. Y-axis label: Change “prot” with “protein”.
Line 188: The representation of enzyme activity in the test is quite confusing. It would be easier if you can express enzyme activity levels in fold change in treatments compared to control.
Line 222-235: The test is incomprehensible. I advise the authors to make shorter sentences and express one idea/result for the sentence.
Fig. 3 Y-axis label: Change “weight gain (mg)” with “larval weight gain (mg)”.
Fig. 3: Both the uppercase and lowercase letters on the top of the bars is not clear and does not show which one is significant compared to what? The authors need to explain or use different letters to make it clear.
The discussion is superficial at this moment and needs improvement. The comparison with other studies and drawing conclusions could improve.
I tried reading many times to understand what the authors trying to say, but I could not make anything out of the test because of so many typos, grammatical errors, and incomprehensible test. Hence, I cannot be able to judge the technicality of this study and I have to give up. Proofreading this manuscript by a native English speaker may improve this manuscript substantially.
Author Response
Reviewer #3:
Point 1:Line 10: The full form of abbreviations needs to be provided when they are first used.
Response 1 :I have already used the full name for the first time.
Point 2:Line 17: Xanthotoxin is written as xanthoxytoxin; correct it.
Response 2 :Thanks, we have modified.
Point 3:Line No 19: Change “by increase” with “by increasing”.
Response 3 :Thanks, we have modified.
Point 4:Line 77: Change “detoxify ability” with “detoxification ability”.
Response 4 :Thanks, we have modified.
Point 5:Line 87: Delete “preparation” in this sentence.
Response 5 :Thanks, we have modified.
Point 6:Line 91: Delete “supplemented” in this sentence.
Materials and Methods should be written in the past tense as they have already been done.
Response 6 :This content has been deleted
Point 7:Line 105: Dissolved xanthotoxin (≥98%, Yuanye, Shanghai, China) by dimethyl sulfoxide. In this sentence change “by” with “in.
Response 7 :Thanks, we have modified.
Point 8:Fig. 1. Y-axis label: Change “prot” with “protein”.
Response 8 :Thanks, we have modified.
Point 9:Line 188: The representation of enzyme activity in the test is quite confusing. It would be easier if you can express enzyme activity levels in fold change in treatments compared to control.
Response 9 :Thanks, we have modified.
Point 10:Line 222-235: The test is incomprehensible. I advise the authors to make shorter sentences and express one idea/result for the sentence.
Response 10 :What I understand is to summarize in the first sentence, and to explain the latter content in detail.
e.g Larvae exposed to different concentrations of MeJA showed somewhat alleviation of the toxic effects of the xanthotoxin, especially seen by weight gain and pupare weight (Fig. 3A, B). The weight gain and pupal weights of the larvae pre-exposed to 20 µg/µL MeJA were 42.18% and 8.03% higher, respectively, than those without MeJA exposure when fed on the 0.05% xanthotoxin diet; the weight gain of the larvae pre-exposed to 200 µg/µL MeJA was 22.19% higher than in those without MeJA exposure when fed on the 0.1% xanthotoxin diet (Fig. 3A, B). The pupal weights of the larvae pre-exposed to 20 µg/µL MeJA were 6.88% higher than those without MeJA exposure when fed on the 0.2% xanthotoxin diet (Fig. 3B). the duration of 4th instar and the prepupal instar of the larvae pre-exposed to 20 µg/µL MeJA reduced when the larvae were fed on the 0.05% xanthotoxin diet and the durations of the 5th and 6th instars were unaffected (Table 2).
Point 11:Fig. 3 Y-axis label: Change “weight gain (mg)” with “larval weight gain (mg)”.
Response 11 :Thanks, we have modified.
Point 12:Fig. 3: Both the uppercase and lowercase letters on the top of the bars is not clear and does not show which one is significant compared to what? The authors need to explain or use different letters to make it clear.
Response 12 :The letters on the bar chart have been enlarged, upper letters indicate significant differences in different MeJA concentrations at the same flavotoxin concentration; lower letters indicate significant differences in different flavotoxin concentrations at the same MeJA concentration. Different letters indicate significant differences.
Point 13:The discussion is superficial at this moment and needs improvement. The comparison with other studies and drawing conclusions could improve.
I tried reading many times to understand what the authors trying to say, but I could not make anything out of the test because of so many typos, grammatical errors, and incomprehensible test. Hence, I cannot be able to judge the technicality of this study and I have to give up. Proofreading this manuscript by a native English speaker may improve this manuscript substantially.
Response 13 :Thank the reviewers for the comments,as the reviewer suggested. We have modified the format and language issues. As the reviewer has pointed out, What I understand is to summarize in the first sentence, and to explain the latter content in detail. It has been modified as follows:“Larvae exposed to different concentrations of MeJA showed somewhat alleviation of the toxic effects of the xanthotoxin, especially seen by weight gain and pupare weight (Fig. 3A, B). The weight gain and pupal weights of the larvae pre-exposed to 20 µg/µL MeJA were 42.18% and 8.03% higher, respectively, than those without MeJA exposure when fed on the 0.05% xanthotoxin diet; the weight gain of the larvae pre-exposed to 200 µg/µL MeJA was 22.19% higher than in those without MeJA exposure when fed on the 0.1% xanthotoxin diet (Fig. 3A, B). The pupal weights of the larvae pre-exposed to 20 µg/µL MeJA were 6.88% higher than those without MeJA exposure when fed on the 0.2% xanthotoxin diet (Fig. 3B). the duration of 4th instar and the prepupal instar of the larvae pre-exposed to 20 µg/µL MeJA reduced when the larvae were fed on the 0.05% xanthotoxin diet and the durations of the 5th and 6th instars were unaffected (Table 2). ”
Point 14:Fig. 3: Both the uppercase and lowercase letters on the top of the bars is not clear and does not show which one is significant compared to what? The authors need to explain or use different letters to make it clear.
Response 14 :The letters on the bar chart have been enlarged, upper letters indicate significant differences in different MeJA concentrations at the same flavotoxin concentration; lower letters indicate significant differences in different flavotoxin concentrations at the same MeJA concentration. Different letters indicate significant differences.
Point 15:The discussion is superficial at this moment and needs improvement. The comparison with other studies and drawing conclusions could improve.
Response 15 : As the reviewer has pointed out, the discussion is superficial at this moment and needs improvement, we have conducted an expanded discussion through our conclusions as well as in combination with other studies.
Round 2
Reviewer 3 Report
Comments to authors:
I carefully went over the revised manuscript and the authors have made some improvements in the manuscript presentation. However, the English language (both grammar and sentence structure) is still problematic as it suffers from numerous instances of incorrect grammar, poor word choice, and misspellings. Some specific instances are noted below (among others). I think the manuscript should be published after addressing the problems listed below.
Summary: Change “interbiotic” with “biotic”.
Abstract: Change “induced higher enzyme-detoxification enzyme” with “induced higher detoxification enzymes”
Introduction: Change “provide defense to plants, since they are able to use” with “aid plants defenses against insect pests through”
Materials and Methods:
2.1 Correct spelling error “holine Chloride” with “choline chloride”
2.4 Delete “Ice tray”
Change “detection kit” to “detection kits”
Change “The unit of enzyme activities was U/mg protein” to “The unit of enzyme activities were represented as U/mg protein”.
The authors need to define what is the value of enzyme unit (U).
Did the authors use Coomassie blue method with standards for determining the protein concentration? It is not clear from the test.
Results:
3.1 Change “(Fig. 1C) compared control” with “(Fig. 1C) compared to control”
3.3 Change “Finally, the pupal weights of 20microgram/microliter-MeJA-treated larvae were 10.84% higher than those without MeJA treatment” with “Finally, the pupal weights were 10.84% higher in 20U/µl in MeJA treatment compared to without MeJA treatment”.
Be consistent with the use of units (use U/µl).
Fig. 2. Y-axis label: Change “prot” with “protein”.
Table 1. The authors can provide exact P-values in this table. Otherwise, this table does not provide much useful information. It can be moved to supplementary, or the P-values can be included in figure 3.
I strongly recommend the authors make shorter sentences and express one idea for a sentence. The lengthy sentences (with >20 words) are incomprehensible. I feel the manuscript requires a significant rewriting effort with the help of a native speaker.
Author Response
Reviewer #3:
Comments and Suggestions for Authors
Comments to authors:
I carefully went over the revised manuscript and the authors have made some improvements in the manuscript presentation. However, the English language (both grammar and sentence structure) is still problematic as it suffers from numerous instances of incorrect grammar, poor word choice, and misspellings. Some specific instances are noted below (among others). I think the manuscript should be published after addressing the problems listed below.
Point 1:Summary: Change “interbiotic” with “biotic”.
Response 1 :Thanks, we have modified.
Point 2:Abstract: Change “induced higher enzyme-detoxification enzyme” with “induced higher detoxification enzymes”
Response 2 :Thanks, we have modified.
Point 3:Introduction: Change “provide defense to plants, since they are able to use” with “aid plants defenses against insect pests through”
Response 3 :Thanks, we have modified.
Point 4:Materials and Methods:
2.1 Correct spelling error “holine Chloride” with “choline chloride”
2.4 Delete “Ice tray”
Change “detection kit” to “detection kits”
Change “The unit of enzyme activities was U/mg protein” to “The unit of enzyme activities were represented as U/mg protein”.
The authors need to define what is the value of enzyme unit (U).
Did the authors use Coomassie blue method with standards for determining the protein concentration? It is not clear from the test.
Response 4 :Thank you for correcting our spelling mistakes, we have modified. And for the unclear elaboration in our method we have added in the method 2.4. We measured the protein by using the Coomassie brilliant blue method,This has already been shown in Method 2.4 “The total protein content was measured and calculated by using the Coomassie blue method following kit instructions (Comin Biotech, Suzhou, China) ”
Point 5:Results:
3.1 Change “(Fig. 1C) compared control” with “(Fig. 1C) compared to control”
Thanks, we have modified.
3.3 Change “Finally, the pupal weights of 20microgram/microliter-MeJA-treated larvae were 10.84% higher than those without MeJA treatment” with “Finally, the pupal weights were 10.84% higher in 20U/µl in MeJA treatment compared to without MeJA treatment”.
Be consistent with the use of units (use U/µl).
Thanks, we have modified.
Fig. 2. Y-axis label: Change “prot” with “protein”.
Thanks, we have modified.
Table 1. The authors can provide exact P-values in this table. Otherwise, this table does not provide much useful information. It can be moved to supplementary, or the P-values can be included in figure 3.
Response 5 :Thank the reviewers for the comments,We have modified the format and language issues. For the reviewer's question, I tried to do the following answers: The concentration unit of methyl jasmonate in our manuscript is ug / ul. In Table 1, the P-value is equal to 0, For Table 1, we just wanted to prove the effect of the interaction of methyl jasmonate and xanthotoxin on the growth and development of S. litura, and the results proved that the combined effect of methyl jasmonate, xanthotoxin, methyl jasmonate and xanthotoxin all affected the growth and development of S. litura. And we have placed Table S1 in the Appendix A.
Point 6:I strongly recommend the authors make shorter sentences and express one idea for a sentence. The lengthy sentences (with >20 words) are incomprehensible. I feel the manuscript requires a significant rewriting effort with the help of a native speaker.
Response 6 :Thank you for your recommendation, our article has been revised by Insects journal English editing Company as requirements, and we have now shortened long sentences to clearly express our ideas.